# DiffusionRollout: Uncertainty-Aware Rollout Planning in Long-Horizon PDE Solving

**Seungwoo Yoo***                                           *dreamy1534@kaist.ac.kr*
*School of Computing*
*KAIST*

**Juil Koo***                                                 *63days@kaist.ac.kr*
*School of Computing*
*KAIST*

**Daehyeon Choi***                                 *daehyeonchoi@kaist.ac.kr*
*School of Computing*
*KAIST*

**Minhyuk Sung**                                                 *mhsung@kaist.ac.kr*
*School of Computing*
*KAIST*

**Reviewed on OpenReview:** *https://openreview.net/forum?id=OCzcGOzgzz*

## Abstract

We propose DiffusionRollout, a novel selective rollout planning strategy for autoregressive diffusion models, aimed at mitigating error accumulation in long-horizon predictions of physical systems governed by partial differential equations (PDEs). Building on the recently validated probabilistic approach to PDE solving, we further explore its ability to quantify predictive uncertainty and demonstrate a strong correlation between prediction errors and standard deviations computed over multiple samples—supporting their use as a proxy for the model's predictive confidence. Based on this observation, we introduce a mechanism that adaptively selects step sizes during autoregressive rollouts, improving long-term prediction reliability by reducing the compounding effect of conditioning on inaccurate prior outputs. Extensive evaluation on long-trajectory PDE prediction benchmarks validates the effectiveness of the proposed uncertainty measure and adaptive planning strategy, as evidenced by lower prediction errors and longer predicted trajectories that retain a high correlation with their ground truths.

## 1 Introduction

Solving partial differential equations (PDEs) is fundamental to scientific computing, enabling the modeling and prediction of a wide range of physical systems—from microscopic phenomena like pattern formation driven by chemical reactions and diffusion, to macroscopic dynamics such as ocean currents that influence global climate. While traditional approaches have relied on numerical solvers due to the lack of closed-form solutions, recent advances such as physics-informed neural networks (PINNs) (Raissi et al., 2019; Krishnapriyan et al., 2021) and neural operators (Li et al., 2020; 2021) have opened new possibilities for reducing computational cost and accelerating solution times. Besides, learning-based solvers enable data-driven simulations of real-world systems with unknown governing equations, as they can be trained directly from observational data without requiring prior knowledge of the underlying dynamics. The computational

---

*Equal contribution.

efficiency and flexibility of learning-based PDE solvers have led to their success across diverse domains, including fluid dynamics (Li et al., 2023c; 2024; Wu et al., 2024) and climate modeling (Kurth et al., 2023).

The standard approach to training neural PDE solvers casts the problem as regression, minimizing empirical risk between model predictions and ground truth. Recent work (Huang et al., 2024; Shysheya et al., 2024), however, adopt a generative perspective, framing PDE modeling as conditional generation—sampling plausible future trajectories conditioned on past observations. This shift has enabled neural solvers to tackle more complex tasks, such as inverse problems (Huang et al., 2024) and data assimilation (Shysheya et al., 2024), that conventional regression-based methods struggle to address. Moreover, even in purely regression-based settings, the training objective, originally designed for modeling high-dimensional data distributions in diffusion and score-based generative models (Song & Ermon, 2019; Ho et al., 2020; Song et al., 2021), naturally extends to PDE learning, where discretized solution functions are typically high-dimensional (Hu et al., 2025). More importantly, it mitigates spectral bias—where regression losses overemphasize low-frequency components (Lippe et al., 2023)—which adversely affects the long-term stability of solvers, particularly in nonlinear systems.

In this work, we further explore benefits of generative modeling for PDEs, focusing on the challenging task of long-term prediction through autoregressive rollouts, where the model recursively uses its own past predictions to generate future timesteps. Although this approach has been empirically shown to outperform direct prediction methods (Wang & Perdikaris, 2023; Li et al., 2022)—which parameterize the solver to estimate the system state at a target time from initial and boundary conditions—it also suffers from a key limitation: exposure bias (Arora et al., 2022), where discrepancies between the model's predictions and the training data accumulate over time, leading to compounding errors. This issue becomes particularly pronounced in long-term predictions of physical systems described by continuous functions over space and time, where even small prediction errors can rapidly accumulate and destabilize the solution.

The core idea behind DiffusionRollout that addresses this challenge is to take advantage of the probabilistic nature of diffusion models and incorporate sampling-based uncertainty estimates. Our analysis reveals that statistics computed from multiple samples drawn from a conditional diffusion model—trained to approximate the distribution of possible future trajectories—are strongly correlated with prediction errors. This makes them a reliable metric for estimating the model's confidence in its predictions during rollouts. Based on this, we propose a simple yet effective training-free selective planning strategy, where unreliable future predictions are discarded and the model advances only to timesteps where it is sufficiently confident. In experiments, we task neural PDE solvers with challenging long-trajectory predictions and demonstrate the effectiveness of the proposed framework, which sets a new state-of-the-art over recent neural operators. Moreover, comparisons with the base model and the original rollout strategy—where the sliding window advances by the largest steps possible—clearly show performance gains achieved without any additional training.

## 2  Related Work

**Neural PDE Solvers and Long-Term Rollouts.**  Neural PDE solvers offer several advantages over traditional numerical methods, including modeling of physical systems with unknown governing equations using physics-informed neural networks (PINNs) (Raissi et al., 2019; Krishnapriyan et al., 2021; Cho et al., 2024) or neural operators (Li et al., 2020; 2021), prediction of system dynamics from partial or incomplete observations (Huang et al., 2024), and solving control tasks (Hu et al., 2025). Together with the development of network architectures specialized in surrogate learning (Li et al., 2023c; 2024; 2023b;a; Hao et al., 2023; Xiao et al., 2024; Kitaev et al., 2020; Choromanski et al., 2021; Cao, 2021; Wu et al., 2024; Wang & Wang, 2024; Alkin et al., 2024), there is increasing interest in applying such models to achieve accurate long-term rollouts, often spanning hundreds of timesteps (Lippe et al., 2023; Ruhe et al., 2024; Shysheya et al., 2024). Lippe et al. (2023) adapt the iterative denoising scheme of diffusion models (Song & Ermon, 2019; Ho et al., 2020; Song et al., 2021) to better capture high-frequency components in predicted solutions, which is especially critical for nonlinear PDEs. Similarly, Shysheya et al. (2024) investigate conditioning strategies for diffusion models in PDE learning and introduce an autoregressive sampling scheme for long-horizon predictions. Likewise, Ruhe et al. (2024) propose an alternative diffusion time scheduling method that better captures the varying uncertainty across the prediction horizon. Building on this line of work, we

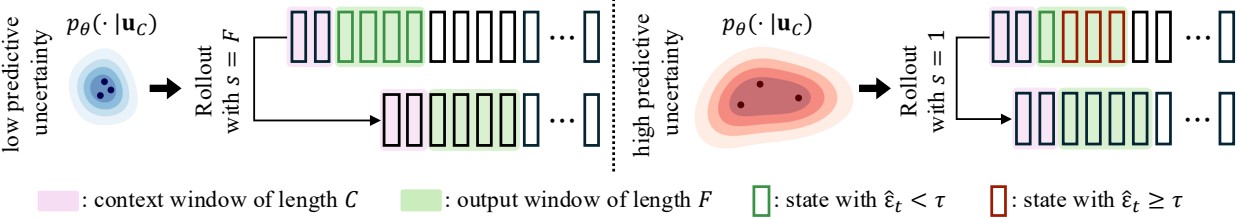

Figure 1: **Adaptive rollout step size $s$ based on predictive uncertainty.** The figure illustrates how predictive uncertainty $\hat{\varepsilon}_t$ governs the rollout step size. On the left, low sample variance leads all future states to satisfy $\hat{\varepsilon}_t < \tau$, allowing a full-step rollout ($s = F$). On the right, high variance (i.e., uncertainty) causes only the first state to pass the threshold, leading to $s = 1$. Based on this, our diffusion-based PDE solver adaptively adjusts the rollout step size during autoregressive prediction.

propose a simple yet effective approach to quantify predictive uncertainty and leverage it to further reduce error during long rollouts.

**Exposure Bias in Autoregressive Generation.** Exposure bias (Ranzato et al., 2016; Bengio et al., 2015; Arora et al., 2022) refers to the discrepancy between training and inference in autoregressive models: during training, the model is conditioned on ground-truth sequences, whereas at inference time, it must rely on its own previous predictions. This mismatch can lead to error accumulation, particularly in long-horizon generation tasks such as language modeling (OpenAI, 2024), visual autoregressive generation (Tian et al., 2024; Han et al., 2024), and PDE solving (Li et al., 2023b; 2021; Wu et al., 2024; Hu et al., 2025; Lippe et al., 2023; Shysheya et al., 2024). To address this issue, a variety of training strategies have been proposed, including adversarial training (Xu et al., 2019), reinforcement learning (Bahdanau et al., 2017; Chen et al., 2020), scheduled sampling or teacher-forcing alternatives (Zhang et al., 2019; Leblond et al., 2018), and unlikelihood-based objectives (Welleck et al., 2020). Despite these advances, most prior work focuses on modifying the training procedure, often requiring models to be retrained from scratch. In contrast, we propose a simple and practical approach to mitigating exposure bias in long-horizon PDE solving without retraining. Our method adaptively adjusts the rollout step size based on the model's prediction uncertainty, balancing a trade-off between network approximation error and inaccurate condition-induced error.

**Uncertainty Quantification Using Diffusion Models.** Uncertainty quantification in learning-based models has been extensively studied as a means of estimating the confidence of model predictions; we refer the reader to the comprehensive survey (Gawlikowski et al., 2023) and provide a brief summary below. Common approaches for estimating uncertainty include aggregating predictions from an ensemble of models (Lakshminarayanan et al., 2017), or training Bayesian neural networks (BNNs) (Depeweg et al., 2018) that model posterior distributions over network weights and make predictions by marginalizing over the learned prior. Similar techniques have only recently been applied to diffusion models (Berry et al., 2024; Chan et al., 2024). Berry et al. (2024) estimate epistemic uncertainty by training multiple submodules of a pre-trained diffusion model and computing mutual information between outputs and weights, whereas Chan et al. (2024) use Bayesian hyper-networks to generate ensembles of model parameters to bypass the need for training multiple networks. However, these approaches inherit the limitations of ensembles and Bayesian neural networks, including high training costs and the need for specialized network designs. In contrast, our method is architecture-agnostic and requires no additional training.

## 3 Long-Horizon Rollouts of PDE Solutions

### 3.1 Learning to Solve Time-Dependent PDEs

We focus on time-dependent partial differential equations (PDEs) defined over a temporal domain $t \in [0, T]$ and a $d$-dimensional spatial domain $\Omega \subseteq \mathbb{R}^d$. In general, the solution $\mathbf{u}(\mathbf{x}, t) : \Omega \times [0, T] \to \mathbb{R}^c$, with temporal

and spatial derivatives satisfies:

$$\frac{\partial \mathbf{u}}{\partial t} = F(\mathbf{u}, \frac{\partial \mathbf{u}}{\partial \mathbf{x}}, \frac{\partial^2 \mathbf{u}}{\partial \mathbf{x}^2}, \cdots) + \mathbf{f}(\mathbf{x}, t), \tag{1}$$

where $F$ denotes a differential operator and $\mathbf{f}(\mathbf{x}, t) : \Omega \times [0, T] \to \mathbb{R}^c$ is a forcing function. Typically, Eqn. 1 is accompanied by the initial condition $\mathbf{u}(\mathbf{x}, 0) = \mathbf{u}_0(\mathbf{x})$ and the boundary condition $B[\mathbf{u}](\mathbf{x}, t) = 0$ for $(\mathbf{x}, t) \in \partial\Omega \times [0, T]$. The main goal of neural surrogate learning is to approximate the solution operator $\mathcal{G} : \mathcal{U} \to \mathcal{U}$ for Eqn. 1 using a neural network $\mathcal{G}_\theta$, parameterized by $\theta$.

Let $\mathbf{u}_{0:T} := [\mathbf{u}(\mathbf{x}, 0), \ldots, \mathbf{u}(\mathbf{x}, T)] \in \mathbb{R}^{(T+1) \times N \times c}$ denote the discretized solution trajectory, obtained by sampling $\mathbf{u}(\mathbf{x}, t)$ at $N$ spatial points in the domain $\Omega$ over $T$ timesteps, each separated by $\Delta t \in \mathbb{R}_{\geq 0}$. When $T$ is small, the entire trajectory can be modeled from the initial condition $\mathbf{u}_0$ using a single forward pass of the model. However, this direct entire trajectory modeling approach has been shown to fall short for large $T$—where $T$ spans tens to hundreds of steps and each temporal solution slice $\mathbf{u}_t$ is a high-dimensional vector. Instead, autoregressive rollouts have proven more effective than modeling the entire trajectory directly at large $T$ (Lippe et al., 2023; Li et al., 2022; Wang & Perdikaris, 2023).

As such, we choose to model short segments of length $T' \ll T$, where each segment is divided into a context of length $C$ and a future portion of length $F = T' - C$, serving as the input and target output of $\mathcal{G}_\theta$, respectively. At inference time, the model $\mathcal{G}_\theta$ predicts long trajectories in an autoregressive manner, using its previous outputs as inputs for the next segment. For training, an entire trajectory $\mathbf{u}_{0:T}$ is now chunked into $M$ pairs of a context and a future portion $\{(\mathbf{u}_{C_i}, \mathbf{u}_{F_i})\}_{i=1}^M$, with $C_i = \{t_i, t_i + 1, \ldots, t_i + C - 1\}$ and $F_i = \{t_i + C, t_i + C + 1, \ldots t_i + T' - 1\}$ indicating the sets of timestep indices belonging to the context and the future portions of the $i$-th pair. Note that we use $C$ and $F$ interchangeably to denote both the index sets and their corresponding cardinalities, for notational convenience. In this setting, the parameters $\theta$ are optimized using these pairs by minimizing the empirical risk:

$$\mathcal{L} = \mathbb{E}_{\mathbf{u}_{0:T} \sim \mathcal{D}} \left[ \sum_{i=1}^M \|\mathbf{u}_{F_i} - \mathcal{G}_\theta(\mathbf{u}_{C_i})\|^2 \right], \tag{2}$$

where $\mathcal{D}$ is the training set. Once trained, the model $\mathcal{G}_\theta$ serves as an approximator of the underlying mapping from context to future, i.e., $\mathbf{u}_F \approx \mathcal{G}_\theta(\mathbf{u}_C)$.

## 3.2 Generative Models in PDE Solving

While regression models trained with Eqn. 2 are commonly used in PDE modeling, recent studies have demonstrated the effectiveness of probabilistic approaches to learning $\mathcal{G}_\theta$. These methods not only enable applications where generative capabilities are essential (e.g., inverse problems) (Huang et al., 2024; Shysheya et al., 2024), but also improve performance in purely predictive tasks (Lippe et al., 2023; Hu et al., 2025). From the perspective of generative modeling, particularly with diffusion or score-based models (Ho et al., 2020; Song et al., 2021), the mapping $\mathbf{u}_F = \mathcal{G}_\theta(\mathbf{C})$ can be viewed as sampling from a conditional distribution. In this framework, a random sample $\boldsymbol{\epsilon} \sim \mathcal{N}(\mathbf{0}, \mathbf{I})$ is gradually transformed into the target $\mathbf{u}_F$—the future portion of a segment—that is plausible given the context $\mathbf{u}_C$:

$$\mathbf{u}_F \sim p_\theta(\mathbf{u}_F | \mathbf{u}_C) \tag{3}$$

where $p_\theta(\mathbf{u}_F | \mathbf{u}_C)$ is a conditional distribution of possible $\mathbf{u}_F$'s. Given sufficient past context $\mathbf{u}_C$, $p_\theta(\mathbf{u}_F | \mathbf{u}_C)$ collapses to a narrow region encompassing a set of plausible future states $\mathbf{u}_F$ with only minor variations. These can be sampled through an iterative denoising or refinement process, where $\mathbf{u}_F$ is progressively reconstructed by recovering both low- and high-frequency components while removing noises of decreasing magnitudes from the initial sample $\boldsymbol{\epsilon}$. Compared to neural solvers trained with regression-based losses, such as the one illustrated in Eqn. 2, this approach has been shown to better capture a broader range of frequencies in the target signal $\mathbf{u}_F$—a key factor in stabilizing long rollouts for nonlinear systems (Lippe et al., 2023). In the following section, we investigate how this probabilistic interpretation naturally enables the quantification of the solver's prediction uncertainty or confidence. We then propose a planning strategy that improves prediction accuracy through a minimal modification to the rollout procedure, without requiring specialized training strategies or changes to the model architecture.

Figure 2: **Analysis of predictive uncertainty $\hat{\varepsilon}_t$ as a surrogate for prediction error.** The proposed quantity exhibits a strong correlation with the true prediction error across multiple PDE benchmarks, including (from left to right) Gray-Scott (Ohana et al., 2024), Turbulent Flow (Ohana et al., 2024), Cahn-Hilliard (Soares et al., 2023), and Anisotropic Diffusion (Koehler et al., 2024). Best viewed when zoomed in.

### 3.3 Adaptive Rollout Step Size via Uncertainty Estimation

When adopting the $C$-in, $F$-out autoregressive scheme discussed in Sec. 3.1, addressing error accumulation is critical for ensuring long-horizon stability. We identify two primary sources of such accumulation: (1) **network approximation error**, which arises from the inherent limitations of the learned model $\mathcal{G}_\theta$ in approximating the true solution operator and tends to increase with the number of network function evaluations (NFEs); and (2) **condition-induced error**, which stems from using inaccurate or uncertain predicted states as context in subsequent steps. These two sources are closely interrelated: a prediction error caused by approximation inaccuracies is likely to propagate as a condition-induced error in subsequent steps, compounding over time and destabilizing the entire trajectory (Ranzato et al., 2016; Bengio et al., 2015; Arora et al., 2022).

Empirically, we observe that across the model output $\mathbf{u}_F$, the prediction error tends to increase with temporal distance from the context $\mathbf{u}_C$. We attribute this trend not only to the inherent difficulty of modeling long-range dependencies, but also to the fact that temporally adjacent frames are more frequently included together in training segments, following the training procedure described in Sec. 3.1. This observation implies that $F$ predictions within the same time window covered by $\mathcal{G}_\theta$ should *not* be trusted equally, and that overall model performance can be improved by selectively using only the more reliable predictions as context for subsequent steps.

Specifically, let $s \in \{1, 2, \ldots, F\}$ denote the step size for selecting predictions from $\mathbf{u}_F$. A straightforward strategy to mitigate condition-induced error is to follow the most conservative policy by setting $s = 1$, accepting only a single prediction from $\mathbf{u}_F$, discarding the remaining $F-1$ predictions, and advancing the time window by just one step. While this reduces reliance on potentially inaccurate predictions, it leads to a higher number of network function evaluations (NFEs), thereby increasing the likelihood of accumulating network approximation error. On the other hand, using the largest step size $s = F$ accepts all $F$ predictions and uses the last $C$ of them as context for the next rollout. This reduces the number of NFEs and, consequently, the approximation error. However, it increases reliance on less accurate predictions, which amplifies condition-induced error.

Therefore, an ideal way to suppress error accumulation is to identify which type of error—network approximation error or condition-induced error—is more dominant and adapt the rollout step size accordingly. Inspired by adaptive step size methods in numerical analysis (Runge, 1895; Kutta, 1901; Hairer et al., 1987), we propose a simple yet effective adaptive rollout step size strategy.

However, the true prediction error required to determine the rollout step size is inaccessible at inference time. While conventional adaptive step size methods address this limitation by estimating error through high-order approximations, such estimates are often computationally expensive and sensitive to numerical instability. To this end, we propose an surrogate measure that estimates model confidence and enables the rejection of unreliable outputs during autoregressive rollouts.

Compared to deterministic regression models, a key advantage of modeling a PDE solver as a generative model is its ability to estimate uncertainty through sampling. Predictions for uncertain future timesteps

tend to vary across different samples drawn from $p_\theta(\mathbf{u}_F \mid \mathbf{u}_C)$, providing a natural measure of uncertainty. We leverage this by computing statistical measures, specifically, standard deviation, over these samples as a surrogate for the model's predictive uncertainty. Since the underlying PDE dynamics in Eqn. 1 considered in our work are deterministic and the spatial discretization is sufficiently fine, the dominant source of predictive uncertainty is *epistemic*, arising from model approximation and limited context. Our sample-based standard deviation therefore provides a natural measure of this epistemic uncertainty in the learned generative surrogate.

Formally, for each $t \in \{1, 2, \ldots, F\}$, the predicted state $\mathbf{u}_t \in \mathbb{R}^{N \times c}$ refers to the $t$-th slice of the future states $\mathbf{u}_F \in \mathbb{R}^{F \times N \times c}$. The uncertainty $\hat{\varepsilon}_t$ of $\mathbf{u}_t$ is defined as the norm of the element-wise standard deviation computed from a set of samples $\{\mathbf{u}_t^{(k)}\}_{k=1}^K$, where $K$ is the number of samples drawn from the conditional distribution $p_\theta(\cdot \mid \mathbf{u}_C)$:

$$\hat{\varepsilon}_t = \|\sigma\left(\{\mathbf{u}_t^{(k)}\}_{k=1}^K\right)\|_2, \tag{4}$$

where $\sigma\left(\{\mathbf{u}_t^{(k)}\}_{k=1}^K\right) \in \mathbb{R}^{N \times c}$ denotes the element-wise standard deviation of the $K$ samples, and $\|\cdot\|_2$ is the Euclidean norm.

Across all four benchmark PDEs used in our experiments in Sec. 4, the proposed uncertainty measure $\hat{\varepsilon}_t$ shows a strong correlation $(\rho)$ with the actual prediction error, as illustrated in Fig. 2 together with the corresponding fitted lines. This supports the use of uncertainty as a surrogate for prediction error. Leveraging this, at each prediction step, we compare the uncertainty of each predicted state to a predefined tolerance $\tau$, and accept only the states whose uncertainty falls below $\tau$ for use as context in the subsequent rollout. The rollout step size $s$ is then selected as :

$$s = \max(\{t \in \{1, 2, \ldots, F\} | \hat{\varepsilon}_t < \tau\} \cup \{1\}). \tag{5}$$

## 4 Experiments

### 4.1 Experiment Setup

We evaluate the effectiveness of our method across various PDE types through both qualitative and quantitative comparisons against state-of-the-art neural PDE solvers.

**Baselines.** We compare DiffusionRollout against state-of-the-art neural operators, including regression-based methods—FNO (Li et al., 2021), FactFormer (Li et al., 2023b), and Transolver (Wu et al., 2024)—as well as the diffusion-based neural operators—PDE-Refiner (Lippe et al., 2023) and WDNO (Hu et al., 2025). For a fair comparison, all models are trained to predict six future states from four past observations, with parameter counts standardized to approximately 3 million, except for WDNO (Hu et al., 2025), which uses 21 million parameters due to convergence issues. Furthermore, for PDE-Refiner (Lippe et al., 2023), which uses constant rollout step sizes $(s = 1, \ldots, 6)$ and serves as our baseline, we adopt the same network architecture and weights as ours. This ensures a fair comparison and allows us to isolate and evaluate the effectiveness of the proposed adaptive stepping technique described in Sec. 3.3, which selectively retains predicted future states during inference.

**Datasets.** We benchmark on challenging *long*-trajectory simulations of length $T \approx 100$, where autoregressive rollout is crucial for inference. Unless otherwise noted, all simulations are conducted on a unit square discretized at a $64 \times 64$ resolution. Our evaluation spans diverse PDE systems—ranging from reaction-diffusion-driven pattern formations to turbulent flow, as detailed below. Across all PDE systems, we assume the underlying governing equations are *unknown* and incorporate no prior knowledge during training or inference.

1. **Gray-Scott Reaction-Diffusion Equation**: Gray-Scott Reaction-Diffusion models the evolution of the concentrations $X(\mathbf{x}, t)$ and $Y(\mathbf{x}, t)$ of two chemical species, undergoing simultaneous reaction and

diffusion:

$$\frac{\partial X}{\partial t} = \delta_X \Delta X - XY^2 + f(1-X), \quad \frac{\partial Y}{\partial t} = \delta_Y \Delta Y - XY^2 - (f+k)Y,$$

where $\delta_X$ and $\delta_Y$ are the diffusion coefficients, and $f$ and $k$ denote the feed and kill rates, respectively. The concentrations $X(\mathbf{x}, t)$ and $Y(\mathbf{x}, t)$, evolving over space and time, produce complex patterns that are highly sensitive to the parameters $f$ and $k$. This sensitivity makes the problem an ideal testbed, as diverse and intricate trajectories can be generated by varying these values. We use 1,200 trajectories from `The Well` dataset (Ohana et al., 2024), following the provided train-test split and subsampling every 10 frames to obtain sequences of length 101. At test time, all models are tasked with predicting 97 future states based on observations from the first four timesteps of each simulation.

2. **2D Turbulent Radiative Flow Equation**: In this setup, we consider a turbulent flow simulation as a benchmark problem, which models turbulent interaction between two gases—one cold and one hot—initially moving relative to each other. As they gradually mix over time, they form a highly reactive intermediate-temperature gas. This behavior of the system is modeled by the equation:

$$\frac{\partial \rho}{\partial t} + \nabla \cdot (\rho v) = 0, \quad \frac{\partial \rho v}{\partial t} + \nabla \cdot (\rho vv + P) = 0,$$
$$\frac{\partial E}{\partial t} + \nabla \cdot ((E+P)v) = -\frac{E}{t_{\text{cool}}}, \quad E = \frac{P}{\gamma - 1},$$

where $\gamma = \frac{5}{3}$, with $\rho$ denoting the density, $v$ the velocity, $P$ the pressure, $E$ the total energy, and $t_{\text{cool}}$ the cooling time. Due to the highly stochastic nature of gas mixing, accurately predicting future states during rollouts becomes challenging, making this setup suitable for our experiments. As with the Gray-Scott reaction-diffusion system, we use 90 trajectories of length 101 from the `The Well` dataset (Ohana et al., 2024; Fielding et al., 2020), following the same training and test splits. All models use the first seven timesteps as context and are tasked with predicting the remaining 94 future states.

3. **Cahn-Hilliard Equation**: Cahn–Hilliard equation models phase separation in binary mixtures, capturing how two components—such as oil and water—gradually separate into distinct regions over time. Letting $c(\mathbf{x}, t)$ denote the concentration of one component, the equation is given by:

$$\frac{\partial c}{\partial t} = M \left( -\kappa \nabla^4 c + \nabla^2 f'(c) \right),$$

where $M$ denotes the mobility, and $f(c) = Wc^2(1-c)^2$ is the bulk free energy density, with $W$ representing the height of the thermodynamic barrier. The trajectories exhibit continuously evolving component concentrations, with dynamic patterns that emerge and fade over time. Using a numerical solver (Soares et al., 2023), we generate 960 training trajectories and 120 test trajectories, each consisting of simulation results over 101 timesteps. During inference, models are given the first seven timesteps as context and tasked with predicting the remaining sequence.

4. **Anisotropic Diffusion**: The final problem setup we consider is anisotropic diffusion, a generalized form of the heat equation in which the diffusion coefficient varies across the domain:

$$\frac{\partial u}{\partial t} = \nabla \cdot A \nabla u, \tag{6}$$

where $u$ denotes the evolving scalar field (e.g., temperature or intensity), and $A$ is a spatially varying, typically positive-definite diffusion tensor that controls the local direction and rate of diffusion. To generate data, we use the anisotropic diffusion simulator from APEBench (Koehler et al., 2024), producing a total of 1080 trajectories, which are split into 960 for training and 120 for testing. All simulations are run with the default parameters provided by the benchmark. At inference time, the models predict the remaining 97 frames using the first four frames as context.

**Evaluation Metrics.** Given a ground-truth solution $\hat{\mathbf{u}}$ and a model's corresponding prediction $\mathbf{u}$, we use the relative $\mathcal{L}_2$ error to evaluate the accuracy of model predictions:

$$\text{Rel } \mathcal{L}_2(\mathbf{u}, \hat{\mathbf{u}}) = \frac{\|\mathbf{u} - \hat{\mathbf{u}}\|_2}{\|\hat{\mathbf{u}}\|_2}. \tag{7}$$

Furthermore, following PDE-Refiner (Lippe et al., 2023), to evaluate long-term stability, we additionally compute the per-timestep Pearson correlation coefficient $r(t)$ between the predicted and ground-truth states at each timestep. Specifically, for each time index $t$ and , we compute Pearson correlation $r(t)$ between the predicted state $\mathbf{u}_t$ and ground-truth state $\hat{\mathbf{u}}_t$ as:

$$r(t) = \frac{\text{Cov}(\mathbf{u}_t, \hat{\mathbf{u}}_t)}{\sqrt{\text{Var}(\mathbf{u}_t)\text{Var}(\hat{\mathbf{u}}_t)}}, \tag{8}$$

where $\mathbf{u}_t, \hat{\mathbf{u}}_t \in \mathbb{R}^{N \times c}$ are flattened into vectors prior to computing the correlation. After computing per-timestep Pearson correlations, we quantify long-term stability by measuring the average normalized time up to which the correlation stays above 0.9, denoted as $T^{>0.9}$:

$$T^{>0.9} = \frac{1}{T} \max \{t \in [1, T] \mid r(t) \geq 0.9\}. \tag{9}$$

This metric reflects the average proportion of the trajectory during which the prediction remains highly corrleated with the ground-truth PDE trajectory.

**Implementation Details.** We implement our diffusion-based neural solver, including PDE-Refiner (Lippe et al., 2023), using the DiT (Peebles & Xie, 2023) architecture equipped with 3D self-attention layers and rotary embeddings to jointly capture temporal and spatial dependencies. Rather than designing a separate module to process the conditioning input $\mathbf{u}_C$, we concatenate the full sequence $[\mathbf{u}_C \| \mathbf{u}_F]$ and feed it directly into the 3D self-attention layers, together with an indicator specifying which frames correspond to conditioning inputs. To enable this, we adopt the training objective from Diffusion Forcing (Chen et al., 2024; Song et al., 2025), which assigns an independent noise level to each frame. This simplifies the handling of conditioning frames by assigning them a noise level of zero, while applying nonzero noise levels to all remaining frames. At inference time, we employ DDIM (Song et al., 2022) sampling with 10 steps. For all regression-based neural operators, including FNO Li et al. (2021), FactFormer Li et al. (2023b), and Transolver Wu et al. (2024), we use the implementations provided by Wu et al. (2024).

To compute the standard deviation for uncertainty quantification, we draw two samples ($K = 2$), which can be obtained efficiently via batched inference, incurring no significant computational overhead. We use uncertainty thresholds of 1, 1, 5, and 0.32 for the Gray-Scott Reaction-Diffusion Equation, the Cahn-Hilliard Equation, the 2D Turbulent Radiative Flow Equation, and Anisotropic Diffusion, respectively.

## 4.2 Comparisons

We begin by summarizing the performance of our method and the baselines, measured in terms of relative $\mathcal{L}_2$ error and long-term stability $T^{>0.9}$, in Tab. 1. Across all three long-trajectory benchmarks introduced in Sec. 4.1, DiffusionRollout outperforms state-of-the-art neural operators based on regression losses (Li et al., 2021; 2023b; Wu et al., 2024; Hu et al., 2025), including models based on Fourier layers (Li et al., 2021) and transformer architectures tailored for operator learning (Li et al., 2023b; Wu et al., 2024). In addition, DiffusionRollout outperforms recent diffusion-based neural operators (Lippe et al., 2023; Hu et al., 2025), including PDE-Refiner (Lippe et al., 2023), which uses default autoregressive sampling with fixed step sizes, and WDNO (Hu et al., 2025), which leverages wavelet representations to better capture abrupt changes in solutions. We observe a performance drop in the case of WDNO (Hu et al., 2025), which was originally designed to predict entire short-length trajectories at once. This is due to the accumulation of prediction errors, amplified by repeated encoding and decoding through wavelet transforms during long-horizon rollouts.

While PDE-Refiner (Lippe et al., 2023), used as our base model and employing naive autoregressive sampling with fixed step sizes, achieves performance comparable to or better than other baselines, further improvements are possible by adopting our adaptive sampling technique described in Sec. 3.3. In particular, we

| | Gray-Scott (Ohana et al., 2024) | | Turbulent Flow (Ohana et al., 2024) | | Cahn-Hilliard (Soares et al., 2023) | | Anisotropic Diffusion (Koehler et al., 2024) | |
|---|---|---|---|---|---|---|---|---|
| | Rel $\mathcal{L}_2$ ↓ | $T^{>0.9}$ ↑ | Rel $\mathcal{L}_2$ ↓ | $T^{>0.9}$ ↑ | Rel $\mathcal{L}_2$ ↓ | $T^{>0.9}$ ↑ | Rel $\mathcal{L}_2$ ↓ | $T^{>0.9}$ ↑ |
| *Regression-Based Neural Solvers* | | | | | | | | |
| FNO (Li et al., 2021) | 0.652 | 0.042 | 0.419 | 0.526 | 0.450 | 0.256 | 0.035 | **0.998** |
| FactFormer (Li et al., 2023b) | 0.620 | 0.187 | 0.344 | 0.744 | 0.374 | 0.364 | 0.250 | 0.632 |
| Transolver (Wu et al., 2024) | 0.430 | 0.264 | 0.336 | 0.773 | 0.388 | 0.309 | 0.405 | 0.447 |
| *Diffusion-Based Neural Solvers* | | | | | | | | |
| WDNO (Hu et al., 2025) | 0.792 | 0.022 | 0.843 | 0.000 | 0.567 | 0.069 | 1.152 | 0.000 |
| PDE-Refiner (Lippe et al., 2023) ($s=1$) | 0.409 | 0.304 | 0.330 | 0.799 | 0.285 | 0.514 | 0.048 | 0.992 |
| PDE-Refiner (Lippe et al., 2023) ($s=2$) | 0.406 | 0.307 | 0.321 | 0.800 | 0.275 | 0.543 | 0.040 | 0.993 |
| PDE-Refiner (Lippe et al., 2023) ($s=3$) | 0.416 | 0.303 | 0.322 | 0.791 | 0.267 | 0.558 | 0.033 | 0.994 |
| PDE-Refiner (Lippe et al., 2023) ($s=4$) | 0.431 | 0.293 | 0.322 | 0.833 | 0.275 | 0.541 | 0.031 | 0.994 |
| PDE-Refiner (Lippe et al., 2023) ($s=5$) | 0.443 | 0.285 | 0.322 | 0.919 | 0.280 | 0.531 | 0.027 | 0.995 |
| PDE-Refiner (Lippe et al., 2023) ($s=6$) | 0.462 | 0.275 | 0.323 | 0.809 | 0.289 | 0.475 | 0.029 | 0.995 |
| **DiffusionRollout (Ours)** | **0.391** | **0.311** | **0.307** | **0.926** | **0.254** | **0.626** | **0.026** | 0.996 |

Table 1: **Quantitative comparison of long rollouts.** For each dataset, we report the relative $\mathcal{L}_2$ error and the average duration of high correlation ($T^{>0.9}$). DiffusionRollout consistently achieves the lowest error and competitive or superior long-term stability, outperforming state-of-the-art neural solvers and a naive autoregressive sampling strategy across various rollout step sizes. **Bold** indicates the best for each column.

| | Gray-Scott (Ohana et al., 2024) | Turbulent Flow (Ohana et al., 2024) | Cahn-Hilliard (Soares et al., 2023) |
|---|---|---|---|
| Regression Loss Training | 0.479 | 0.849 | 0.270 |
| Derivative Threshold | 0.395 | 0.320 | 0.256 |
| $\hat{\epsilon}_t$ **Threshold** | **0.391** | **0.307** | **0.254** |

Table 2: **Quantitative results from the ablation study.** The proposed method achieves the lowest error, outperforming both its regression-trained variant and the version using an alternative surrogate for predictive uncertainty. **Bold** indicates the best for each column.

observe a notable error reduction on the Gray-Scott dataset (Tab. 1, first column), where our adaptive planning results in 15.37% error reduction compared to the base model's fixed-step sampling strategy, which advances 6 steps per iteration. We highlight that this improvement is achieved purely by modifying the inference procedure based on the analysis in Sec. 3.3, with no changes to the training process.

The trend observed in Tab. 1 is further supported by the qualitative results in Fig. 3, which illustrate the effectiveness of the proposed method in capturing the long-term evolution of dynamical systems. The first column of the figure shows the field at an early timestep, while the second column displays the target field at a much later timestep, typically separated by several dozen simulation steps. Despite the apparent differences between the initial and target states, our model generates accurate predictions. In contrast, even state-of-the-art neural solvers such as Transolver (Wu et al., 2024) struggle in this setting, producing noticeably different patterns—especially on the Gray-Scott and Cahn-Hilliard datasets (fifth and sixth columns)—as reflected by the brighter error maps.

## 4.3 Ablation Study

We validate our design choices by comparing our method against two variants: (1) the same backbone network trained using the regression loss in Eqn. 2 ("Regression Loss Training"), and (2) an alternative adaptive sampling strategy that uses the temporal derivative computed using the predicted future segment as a proxy for predictive uncertainty ("Derivative Threshold"). Qualitative and quantitative comparisons are summarized in Fig. 4 and Tab. 2, respectively. As shown in the first row of Tab. 2, replacing the diffusion-based training objective with the regression loss in Eqn. 2 results in performance degradation. This is especially pronounced on the Gray-Scott dataset, where the error increases by 8.8%p, highlighting the importance of using a generative training loss. Furthermore, we explored an alternative measure of predictive uncertainty motivated by the observation that systems undergoing abrupt changes are inherently more

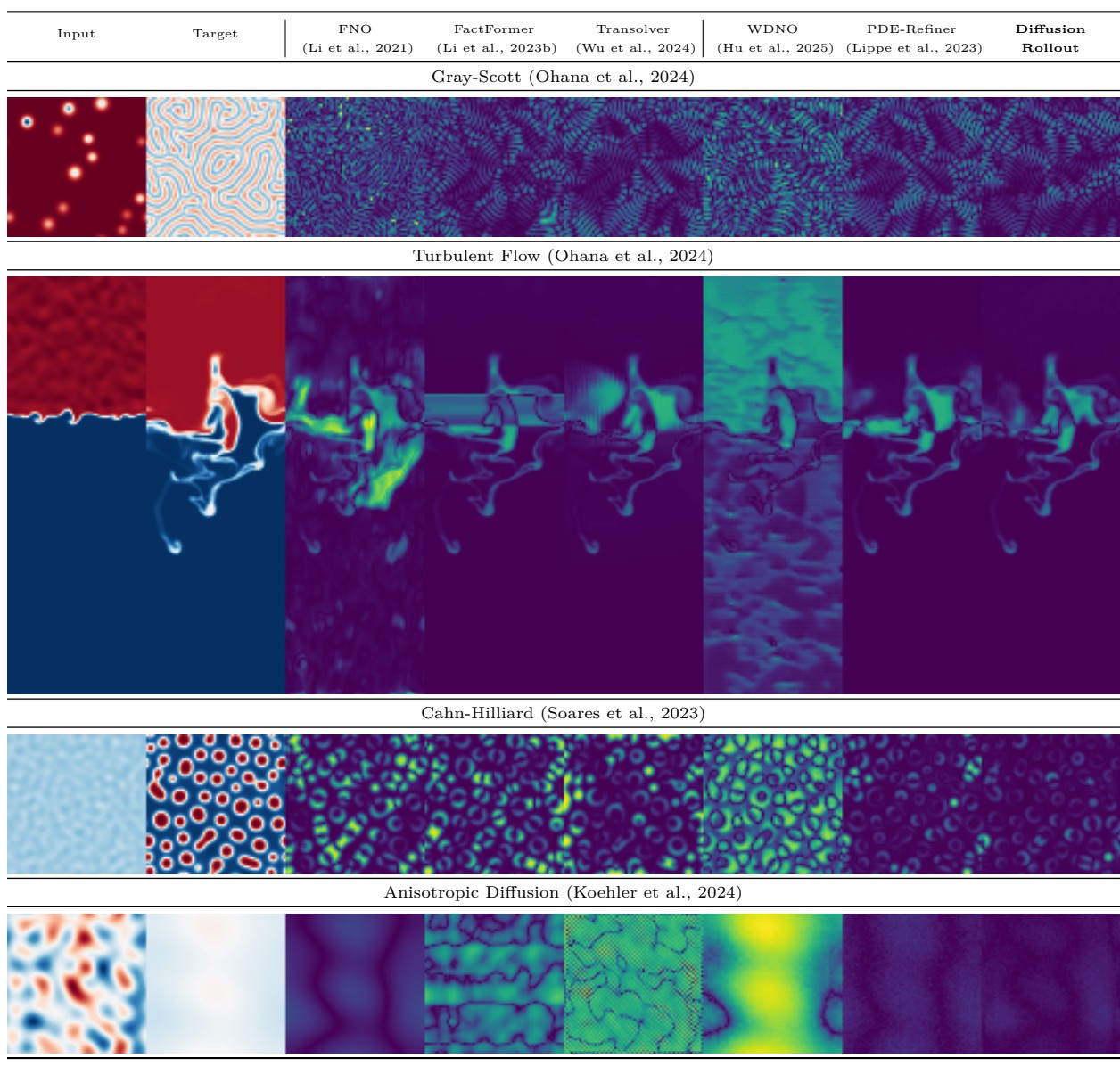

Figure 3: **Qualitative comparison.** Per-pixel error maps (columns 3–8) are visualized for each model's prediction of the target (column 2), given the initial context (column 1). Darker regions in the error maps indicate lower prediction error.

difficult to forecast. Specifically, we consider the temporal derivative of the predicted solution, approximated using the first-order finite difference between consecutive predicted states in $\mathbf{u}_F$:

$$\hat{\varepsilon}_t = \|\mathbf{u}_t - \mathbf{u}_{t-1}\|_2^2. \tag{10}$$

As shown in Tab. 2, computing uncertainty as the standard deviation over multiple samples proves more effective—particularly in the challenging task of predicting turbulent flows.

## 4.4 Additional Analyses

In this section, we present additional analyses, including the impact of various hyperparameter choices on model performance, a runtime comparison against PDE-Refiner (Lippe et al., 2023), and an investigation

| Input | Target | Regression Loss | Derivative Thres. | $\hat{\epsilon}_t$ **Thres. (Ours)** | Input | Target | Regression Loss | Derivative thres. | $\hat{\epsilon}_t$ **Thres. (Ours)** |
|---|---|---|---|---|---|---|---|---|---|
| | | Gray-Scott (Ohana et al., 2024) | | | | | Cahn-Hilliard (Soares et al., 2023) | | |

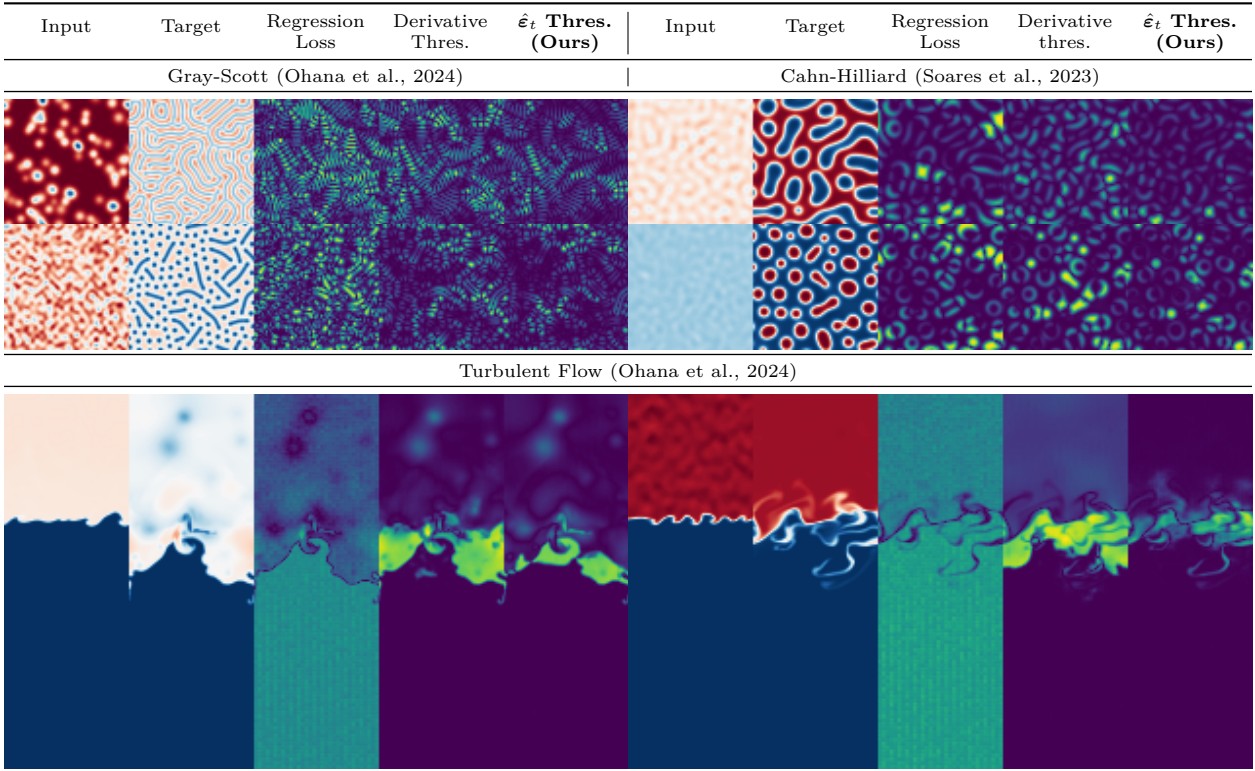

Turbulent Flow (Ohana et al., 2024)

Figure 4: **Qualitative comparison from the ablation study.** Per-pixel error maps (columns 3-5 and 8-10) are shown for each variant of our method, visualizing the prediction error (columns 2 and 7), given the initial context (columns 1 and 6). Darker regions in the error maps indicate lower prediction error.

of the impact of varying rollout step sizes in regression-based baselines (Li et al., 2021; 2023b; Wu et al., 2024). Unless otherwise noted, the following analyses are conducted on the test set of the 2D Gray–Scott dataset (Ohana et al., 2024). We begin by examining the time–accuracy trade-off induced by the choice of uncertainty threshold $\tau$. We then demonstrate that the proposed uncertainty estimates, based on standard deviations, remain reliable even with a small number of samples $K$, and we analyze how the number of diffusion timesteps $S$ affects model accuracy. Finally, we show that, assuming all samples fit into GPU memory, the proposed method incurs only minimal additional overhead compared to PDE-Refiner (Lippe et al., 2023).

**Impact of Varying $\tau$.** We first explore threshold values $\tau \in \{0.001, 0.01, 0.2, 0.6, 3, 5, 7, 9\}$ other than the default value $\tau = 1$ used for the dataset. We summarize the relative $\mathcal{L}_2$ errors corresponding to the step sizes selected by our method for each threshold in Fig. 5. For reference, we also plot the step sizes and predictive errors of PDE-Refiner evaluated at different fixed step sizes. As shown, even with the same model, performance varies with the choice of threshold, highlighting the importance of selecting an appropriate value. Specifically, smaller thresholds make the model more conservative, resulting in smaller steps as reflected by the average step size $s_{\mathrm{avg}}$ along the trajectories.

**Impact of Varying $K$.** Beyond the default number of samples $K = 2$ used throughout the experiments of Sec. 4.2, we additionally evaluate $K \in \{3, 4, 5\}$ to verify how many samples are required for reliable uncertainty estimation. The effect of varying $K$ on prediction accuracy is summarized in Tab. 3. As shown, the relative $\mathcal{L}_2$ error remains consistent across all values of $K$. In particular, $K = 2$ and $K = 5$ yield nearly identical results, suggesting that only a few samples is sufficient for reliable uncertainty estimates.

**Impact of Varying Diffusion Timesteps.** In our experiments, we employ the DDIM sampler (Song et al., 2022) with $S = 10$ timesteps to generate trajectories. To further examine the effect of the number of

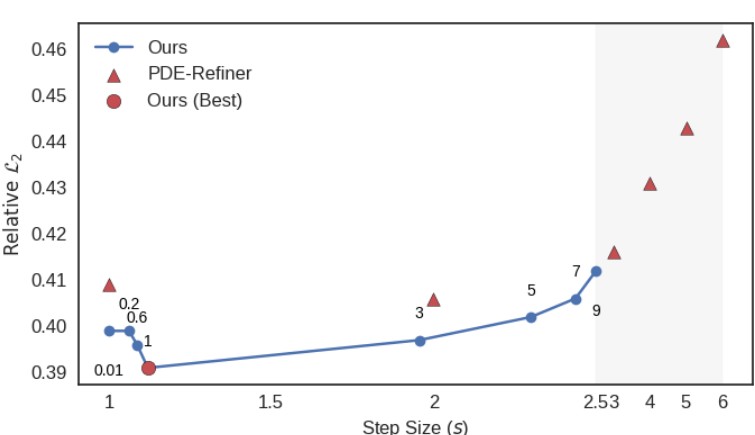

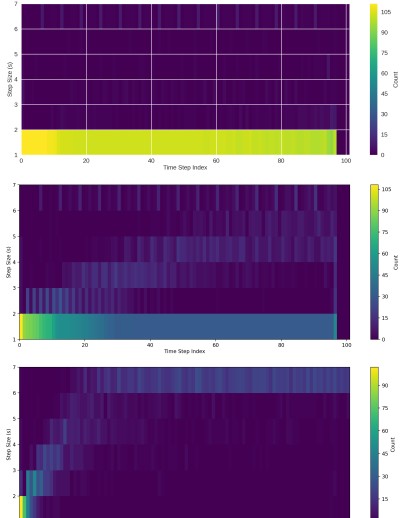

Figure 5: **Relative $\mathcal{L}_2$ errors of predicted solutions versus the average step sizes $s_{\mathbf{avg}}$ under different uncertainty thresholds $\tau$ (shown as overlaid text).** For comparison, the relative $\mathcal{L}_2$ errors of PDE-Refiner with constant step sizes are also visualized. As shown, selecting an appropriate threshold is crucial for balancing prediction accuracy and inference time, as reflected by both the error and the average step size.

Figure 6: **Histograms of step sizes selected along trajectories.** Results are shown for the Gray–Scott equation with uncertainty thresholds $\tau = 1$ (top) and $\tau = 3$ (middle), and for the Cahn–Hilliard equation (bottom).

sampling steps on model performance, we also evaluate the model with larger values $S \in \{20, 30, 40, 50\}$. As shown in Tab. 4, increasing the number of diffusion timesteps and, consequently, NFEs does not necessarily improve accuracy; instead, it slightly increases prediction error, aligning with our discussion in Sec. 3.3.

**Runtime Analysis.** Since our method builds directly on PDE-Refiner (Lippe et al., 2023), its forward-pass computational cost is identical to that of PDE-Refiner, provided that all particles used for uncertainty estimation fit into GPU memory and can be processed within a single batch. Accordingly, we measure both the network forward-pass time for PDE-Refiner (Lippe et al., 2023) and our method, as well as the additional time required to adaptively determine the step size. We report the network forward-pass time and the adaptive step-size selection time based on our uncertainty estimates in Tab. 5. Compared to PDE-Refiner (Lippe et al., 2023) with $s = 1$, whose step size is most similar to that selected by our method, the overall runtime remains comparable, while our approach achieves higher predictive accuracy as shown in Tab. 1. As indicated in the "Adaptive Step Selection (s)" column, computing standard deviations across $K$ samples does not introduce a significant bottleneck.

**Selected Step Size Analysis.** To better understand the behavior of our framework, we track the step size selected at each timestep for every test-set trajectory of Gray-Scott and Cahn-Hilliard equations, and visualize the distribution of these values as a series of histograms. Specifically, we collect the step sizes selected in the Gray-Scott experiments with $\tau = 1$ (best) and $\tau = 3$, as well as in the Cahn-Hilliard experiment, to examine how the selected step sizes vary with the threshold $\tau$ and across datasets. As shown in Fig. 6, the selected step sizes vary noticeably with the choice of $\tau$ and across datasets. On the Gray–Scott dataset with $\tau = 1$ (Fig. 6, top), consistently taking smaller steps leads to improved performance, as also reflected in the PDE-Refiner (Lippe et al., 2023) baselines, and our method automatically discovers this strategy. With the increased uncertainty threshold of $\tau = 3$ (Fig. 6, middle), our method initially behaves conservatively and then gradually increases the step size as the dynamics stabilize. On the other hand, for the Cahn-Hilliard equation, where larger rollout steps are beneficial (as demonstrated by PDE-Refiner (Lippe et al., 2023) with $s = 5$), the proposed framework successfully adapts and selects larger steps accordingly (Fig. 6, bottom).

| $K$ | 2 | 3 | 4 | 5 |
|---|---|---|---|---|
| Rel $\mathcal{L}_2$ | **0.391** | 0.396 | 0.395 | 0.394 |

| $S$ | 10 | 20 | 30 | 40 | 50 |
|---|---|---|---|---|---|
| Rel $\mathcal{L}_2$ | **0.391** | 0.399 | 0.401 | 0.401 | 0.402 |

Table 3: **Relative $\mathcal{L}_2$ errors of predicted solutions for varying sample counts $K$.** Reliable uncertainty estimates can be obtained with as few as two samples.

Table 4: **Relative $\mathcal{L}_2$ errors of predicted solutions for different diffusion timesteps.** Increasing the diffusion timesteps during generation, and thus the NFEs, results in higher errors.

| Method | Network Forward (s) | Adaptive Step Selection (s) | Total (s) |
|---|---|---|---|
| PDE-Refiner (Lippe et al., 2023) ($s = 1$) | 554.623 | – | 554.623 |
| PDE-Refiner (Lippe et al., 2023) ($s = 2$) | 289.360 | – | 289.360 |
| PDE-Refiner (Lippe et al., 2023) ($s = 3$) | 202.421 | – | 202.421 |
| PDE-Refiner (Lippe et al., 2023) ($s = 4$) | 154.558 | – | 154.558 |
| PDE-Refiner (Lippe et al., 2023) ($s = 5$) | 126.524 | – | 126.524 |
| PDE-Refiner (Lippe et al., 2023) ($s = 6$) | 105.005 | – | 105.005 |
| DiffusionRollout (Ours) | 503.251 | 4.250 | 507.501 |

Table 5: **Runtime breakdown of PDE-Refiner and DiffusionRollout.** The proposed method incurs only minimal computational overhead for adaptive step selection. In the experiment on Gray–Scott dataset, where our method uses an average step size of $s_{\mathrm{avg}} = 1.12$, the overall runtime remains comparable to that of PDE-Refiner with a similar fixed step size ($s = 1$).

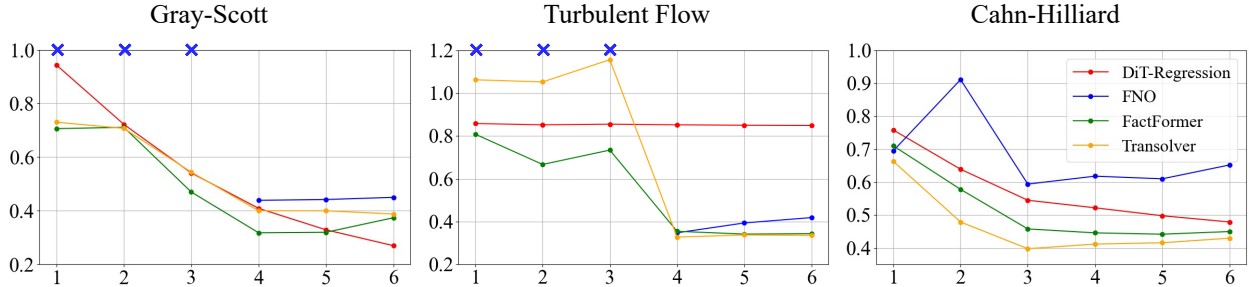

Figure 7: **Relative $\mathcal{L}_2$ errors of regression-based models with varying autoregressive step sizes.** The predictive error generally increases as the step size decreases, indicating that network approximation error is a primary source of error accumulation.

**Varying Step Sizes with Regression-Based Neural Operators.** In addition to our main experiments, which use fixed step sizes for regression-based neural operators (Li et al., 2021; 2023b; Wu et al., 2024), we further examine the effect of varying step sizes in these methods. As shown in the trends in Fig. 7, the relative error generally increases as the step size decreases. We attribute this behavior to regression-based models being more susceptible to network approximation error, where a larger number of NFEs leads to increased error accumulation over long-horizon rollouts. Notably, the trajectories predicted by FNO (Li et al., 2021) diverge and produce invalid numerical values when smaller step sizes are used, as indicated by × markers in the plots. Nevertheless, as discussed in Sec. 4.2, regression-based methods already underperform relative to diffusion-based approaches, and our adaptive rollout strategy is applicable only to the latter.

## 5 Conclusion

We proposed DiffusionRollout, a diffusion-based neural PDE solver designed to predict long rollouts of physical systems governed by PDEs. Building on the benefits of generative modeling for PDE modeling, we introduce a method that leverages the norm of standard deviations—computed from multiple samples drawn from a pre-trained diffusion model—as a surrogate measure of predictive uncertainty. This uncertainty estimate is then used during the autoregressive sampling process to reject erroneous predictions that could otherwise propagate and significantly degrade rollout accuracy. In experiments on three challenging

benchmark problems, the proposed method outperforms state-of-the-art baselines and demonstrates clear performance gains over conventional autoregressive sampling, which, by contrast, does not account for potential prediction errors during inference.

**Limitations and Future Work.** While DiffusionRollout improves long-term prediction accuracy and inherits key benefits of generative modeling, it relies on an iterative sampling process, which is slower than regression-based neural operators. A promising direction for future work is to integrate recent one-step or few-step diffusion models into our adative rollout framework for improved inference efficiency. In addition, although our work primarily focuses on addressing exposure bias along the temporal dimension, a similar issue also arises during the diffusion sampling process due to its iterative nature. We believe this represents another important axis that merits further investigation. Finally, an interesting future direction is to explore learning a dedicated planner module that adapts the uncertainty threshold based on the current context.

## Acknowledgments

This work was supported by the NRF of Korea grant (RS-2023-00209723); IITP grants (RS-2024-00399817, RS-2025-25441313, RS-2025-25443318, RS-2025-02653113); and the Industrial Technology Innovation Program (RS-2025-02317326), all funded by the Korean government (MSIT and MOTIE), as well as grants from the DRB-KAIST SketchTheFuture Research Center.

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

# Appendix

## A.1 Additional Qualitative Results

We present additional qualitative results in Fig. A8, where DiffusionRollout produces the most accurate predictions, as visually indicated by the darker error maps.

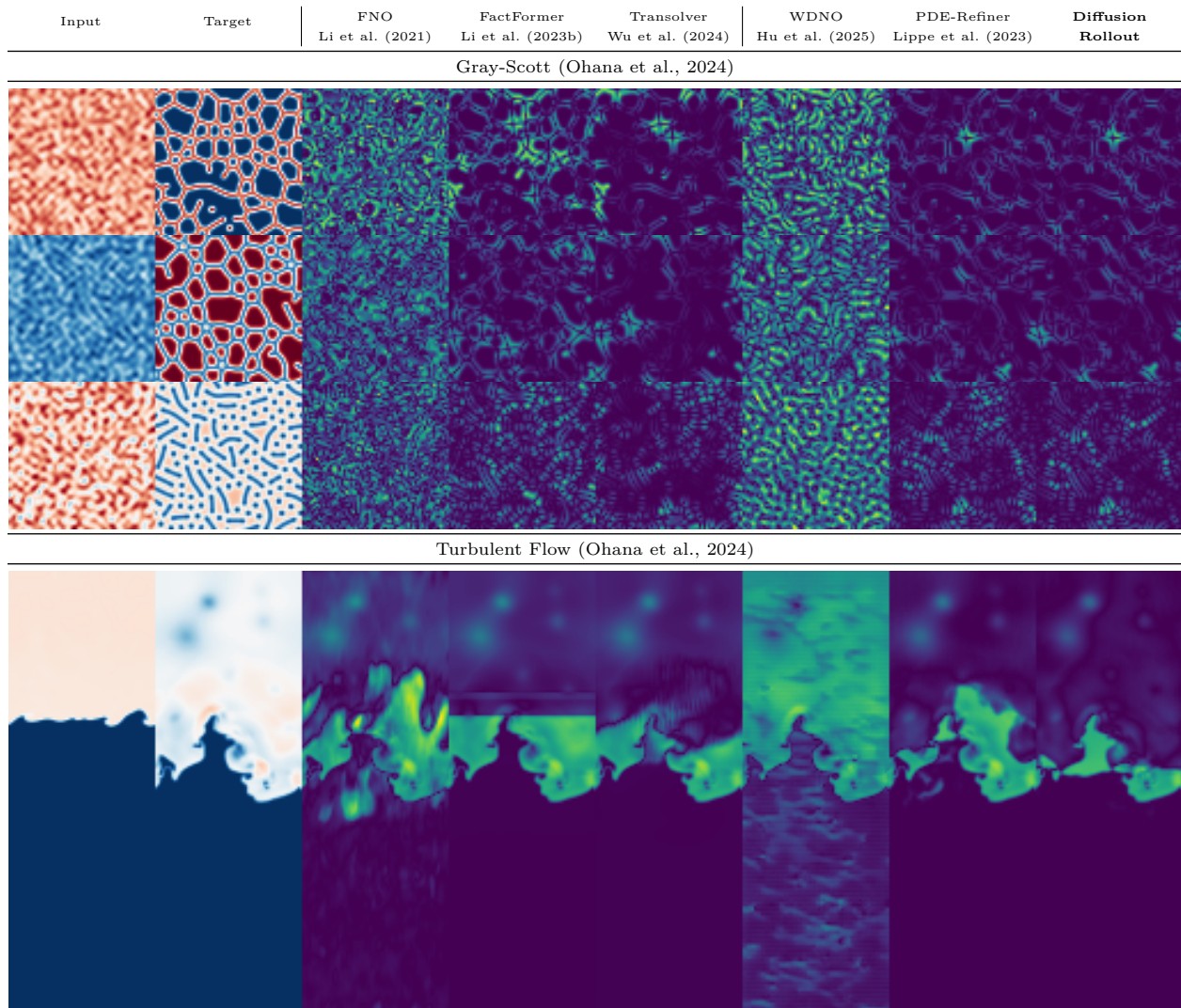

Figure A8: **Additional Qualitative Results.**

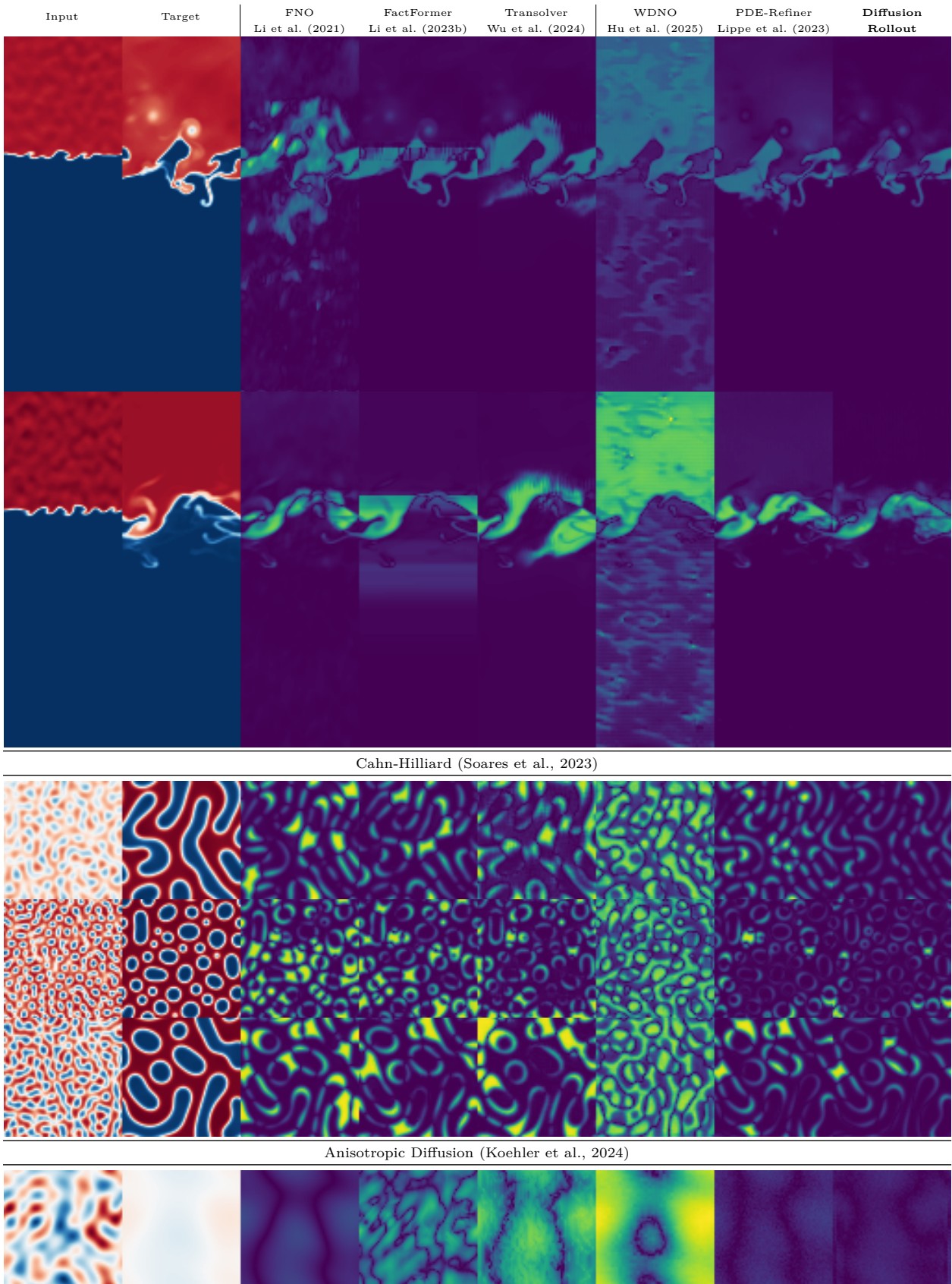

Figure A8: **Additional Qualitative Results.**

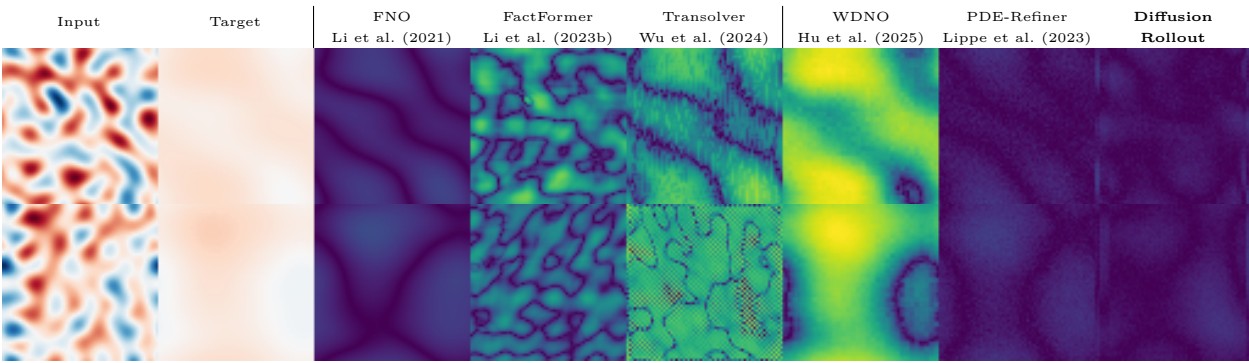

Figure A8: **Additional Qualitative Results.**

