# OpenReview forum: "DiffusionRollout: Uncertainty-Aware Rollout Planning in Long-Horizon PDE Solving"
_TMLR — Accepted by TMLR_

### Review · Reviewer_9oSw · 2025-11-07

**Summary Of Contributions:**

The contributions are:
- The authors propose the standard deviation of a conditional diffusion model to be the surrogate for model's predictive uncertainty and demonstrates a linear relationship between model predictive error and uncertainty on Gray-Scott ReactionDiffusion Equation dataset.
- With the above observation, the authors propose DiffusionRollout, a strategy that adaptively select rollout step size for an autoregressive conditional diffusion model.
- With extensive experiments, the authors demonstrate their strategy helps to mitigate the error accumulation effect over time.

Key strengths:
- The framework is easy to follow.
- Most of the claims are backed by extensive experiments.

Weaknesses:
- While I find the method intuitively correct and backed with numerous experiments, I find it lack some theoretical grounds, without which it is more of an inference heuristic.

**Additional Comments:**

As described above, I would hope more theoretical analysis is available for this framework to be sound and theoretically grounded rather than a heuristic.

**Audience:**

Yes

**Audience Explanation:**

This paper provides a simple training-free inference strategy to adaptively select step size for modeling a PDE. This would be interesting to the operator learning audience.

**Broader Impact Concerns:**

This is no ethical concern.

**Claims And Evidence:**

Yes

**Claims Explanation:**

The claims are supported by evidence. For example:

**Claim**: DiffusionRollout improves long-term prediction reliability.

**Evidence**: DiffusionRollout does best with respect to the $T^{>0.9}$ metric on almost all datasets, indicating that it models long range predictions better than baselines.

However, there is one thing I want to bring up. For the predictive error vs uncertainty relationship (Figure 2), only the Gray-Scott ReactionDiffusion Equation dataset is shown. I wonder if this relationship appears in all datasets.

**Requested Changes:**

1. I would like to see figures like Figure 2 for all other datasets to confirm if predictive error vs uncertainty relationship, a key claim by the authors, is actually universal.
2. The ablation study looks good but to fully evaluate whether the strategy is helpful, I think the following may be useful:
    1. Compare the performance of the model: do naively s=1, s=F and adaptive step size. This is because I find in Table 3, $s_{\text avg}=1.120$ yields the best result, which means the model essentially only do 1 step forward at a time. A histogram of the selected step sizes may also be helpful.
    2. For the above comparison, it is also helpful to show the computational cost for each case, for example NFE, number of FLOPs, etc so we can get a sense of the performance gain and the extra computational overhead.

---

> ### Author Response · Authors · 2025-11-27
> **Author Response**
>
> Dear Reviewer 9oSw,
>
> We sincerely appreciate your thorough review and the insightful comments on our manuscript. We have carefully considered each of your points and revised the manuscript accordingly. Alongside the revised manuscript, we provide the following responses to your comments.
>
> # Predictive Error vs. Uncertainty Plots on Additional Datasets
>
> Thank you for the suggestion. We have included the requested plots in Fig.2 of the revised manuscript.
> We find that, despite its simplicity, the proposed uncertainty measure provides a reliable proxy for the predictive error across diverse datasets.
>
> # Detailed Analysis of Computational Costs
>
> **Comparison with Fixed-Step Baselines.** Thank you for the comment. We apologize for not clearly describing the baseline settings, and we have revised Sec. 4.1 accordingly. In our experiments, we use PDE-Refiner as the base model. Our main contribution is an adaptive step-size selection technique grounded in the proposed uncertainty estimate, which requires no architectural modifications or additional training. Accordingly, the fixed-step PDE-Refiner results correspond to the baselines you requested.
>
> We also agree that a histogram of the selected step sizes would provide additional insight into the behavior of our method, and we have included this in the revised manuscript (see Fig. 6). Specifically, we tracked the step size chosen at each timestep for every trajectory in the test set and visualized the distribution of these per-step values as a series of histograms.
>
> **Quantitative Analysis of Computational Efficiency.** Thank you for the suggestion. As noted in our earlier response on “Comparison with Fixed-Step Baselines”, our method and PDE-Refiner share the same architecture and training procedure. Therefore, we agree that comparing NFEs and providing a runtime breakdown for each method would be informative. For NFEs, we observe that generating $K$ particles for the standard deviation computation can be efficiently batched. Therefore, the NFE remains identical to that of PDE-Refiner, provided that all $K$ particles fit into GPU memory. Furthermore, the time required to compute the uncertainty estimate from the particles is negligible compared to the neural network evaluation cost. We have summarized this in Tab. 5 of the revised manuscript.

---

> > ### Comment · Reviewer_9oSw · 2025-11-28
> >
> > Thanks for the response. I think Fig 2 show a universal trend across dataset, which greatly supports the key claims of this paper. Fig 6 show the model actually select steps adaptively although the average is 1.12 which seems to be insignificant at first glance. Tab 5 also solves my concern on computational cost.
> >
> > Overall, I think this paper is good shape.

---

### Review · Reviewer_rtig · 2025-11-10

**Summary Of Contributions:**

This paper proposes the DiffusionRollout method, whose core idea lies in adaptive step size selection. The authors construct a proxy task based on the prediction error of diffusion models to evaluate the confidence of predictions at different steps of the Solver model. Furthermore, extensive experimental evaluations validate the effectiveness of the method.

**Audience:**

Yes

**Audience Explanation:**

Investigating PDE solvers based on generative models is a meaningful task, and the adaptive step selection explored in this paper is a key factor in the solution process. I believe readers of TMLR will find this topic of interest.

**Claims And Evidence:**

Yes

**Claims Explanation:**

The author effectively validated the core motivation of this paper in Figure 2, demonstrating a positive correlation between the prediction error of the diffusion model and the uncertainty of the Solver model, consistent with the design of the proposed method.

**Requested Changes:**

To my knowledge, the exposure bias issue is not confined to autoregressive generation but is also prevalent in diffusion models. It manifests specifically in samples generated with fewer inference steps (e.g., 300 NFE), outperforming those from full inference (1000 NFE). For instance, DDPM-IP [1] employs input perturbation to mitigate this problem. I believe this is relevant to the methods explored in this paper. The authors should analyze and discuss this issue to provide a more in-depth exploration of exposure bias.

[1] Input Perturbation Reduces Exposure Bias in Diffusion Models. ICML-2023

---

> ### Author Response · Authors · 2025-11-27
> **Author Response**
>
> Dear Reviewer rtig,
>
> We sincerely appreciate your thorough review and the insightful comments on our manuscript. We provide our response in the following.
>
> ## Discussion of Exposure Bias in Diffusion Models
>
> As you correctly pointed out, exposure bias is also a known issue in diffusion sampling, where prediction errors can accumulate due to the iterative nature of the sampling process. We agree that this is an important aspect of modern video diffusion models that merits further investigation. While our work primarily focused on exposure bias along the temporal dimension of generated sequences, we acknowledge the significance of exposure bias in diffusion models and believe that exploring its implications and potential mitigation strategies would be a valuable direction for future research.
> We have updated the manuscript to address this point and have included the discussion in the conclusion.

---

### Review · Reviewer_U8mi · 2025-11-13

**Summary Of Contributions:**

The authors tackle the problem of PDE solving, framing it as conditional generation, and sampled via autoregressive rollouts.
In particular, they stress that although successful, this approach's performance degrades for long horizon since error tends to accumulate leading to what is referred as 'exposure bias' (i.e. distribution shift when chaining the conditional generation).
They also show that the model's variance is predictive of the prediction error, and use this to automatically adjust the rollout window.

**Audience:**

Yes

**Audience Explanation:**

This manuscript is tackling a relevant task since solving PDE is ubiquitous and extremely costly as the resolution or horizon increases.
What's more this manuscript shows that one can adapt the 'time step' depending on the 'model's confidence' to reach a better accuracy-compute tradeoff.

**Claims And Evidence:**

Yes

**Claims Explanation:**

## Strengths
The manuscript is overall pretty easy and pleasant to read.
The introduced sampling procedure is quite simple yet neat.
The suggested method is evaluated on a wide range of datasets.

## Weaknesses
I feel like the motivation for using the conditional's variance as an estimate of prediction error could be strengthen.
Albeit quite impressive, I find the empirical result section a bit sparse in details, and some of the result not obvious to interpret. As such I have written quite a couple of remarks/questions in the hope to get answers and better understand these results.

**Requested Changes:**

- Sec 3.2: I feel like it would be worth saying a bit more about uncertainty. Here there is no aleatoric uncertainty right since we are interested in PDEs there is no 'real' noise. The only uncertainty is thus epistemic. Given $u_s$ a solution of the PDE at time $s$, any future state $u_s$ at time $t>s$ can be _deterministically_ predicted via Eq (1). What breaks this is the space discretisation, and thus $p(u_t|u_s)$ is not fully 'determined' as a such this conditional has non zero variance. That is why conditioning on longer past trajectory narrows the conditional's variance.
- Sec 3.3:
    - Assuming that the space discretisation is not too coarse, the most of the uncertainty is _epistemic_ which is exactly the uncertainty that is aimed to be measured. I would suggest making this connection explicit as it may help justify why this is a 'natural' measure of uncertainty'.
    - Also worth stressing that using the conditionally generated sample's _standard deviation_ as the measure of uncertainty is reasonable because these conditionals are very likely _unimodal_ as they are only conditioned and predicting over a small period of time.
    - I am not totally sure to know/understand in Eq (4) what is the l2 norm for there? What is $\sigma$? Worth clarifying these notations. I would use $\hat{\mathrm{Std}}$. Estimating this Std though necessitate to get several samples though right? Which would not be necessary otherwise, thus somewhat counterbalancing the original motivation of getting a better accuracy/compute tradeoff. Or is the Std estimated across the 6 prediction window states?
- Sec 4.2:
  - I am confused about the difference with PDERefiner, as I perhaps wrongly assumed that this work was building on PDERefiner, but changing the sampling procedure to choose on the fly the prediction window step $s$. If so it feels that the performance of 'DiffusionRollout' should be upper bounded by 'PDERefiner' $(s = 1)$ and lower bounded by 'PDERefiner' $(s = 6)$. I thought that the hope was that this adpaptive step procedure would perform competitively than always choosing $s=1$ but for a fraction of the compute. If 'DiffusionRollout' is _not_ based on 'PDERefiner', then Tab 1 should have results for 'DiffusionRollout' with $s=1,...,6$.
  - Since efficiency is at the core of the motivation for this sampling method it it quite necessary to have some metric measuring this across different models. E.g. is one NFE of PDERefiner the same as one NFE for 'DiffusionRollout'? Are both using the same number of sampling step (per conditional step)?
  - Are the sampling procedure normalised for compute? I assume not since PDERefiner with $s=6$ should be roughly 6 times cheaper than with $s=1$.
- Sec 4.3: 'temporal derivative computed using the predicted future segment' -> Can you say more about this, e.g. write down the formula? Is this looking at the normal of the finite difference estimate of the temporal derivative? Still this method seems quite good too!
- Sec 4.24:
  - Worth making Tab 3 into a figure!
  - Since $s_avg=1.12$ for $\tau=1$ this means that actually it's really not so different then simply choosing $s=1$ right.
  - Thus what's interesting is the hopefully better trade off for higher value of $s_avg$. It would be worth having on that Tab (or Figure) the comparison with the naive approach of always choosing $s=1,..,6$.
  - Also it would be really useful for higher $s_avg$ to see the average value of $s$ for each time along the trajectory. I assume that it starts large ish and then decreases pretty quickly till reaching 1?
  - It is not obvious to me that the optimal performance is for $s_avg>1$ and not $s_avg=s=1$. As in there is a range of $tau$ for which there is no trade off between accuracy and speed, it's a win-win. What is your explanation for this?
  - Tab 4 & 5: Similarly, why would a worse estimate lead to a better choice for $s$?

---

> ### Author Response · Authors · 2025-11-27
> **Author Response**
>
> Dear Reviewer U8mi,
>
> We sincerely appreciate your thorough review and the insightful comments on our manuscript. We have carefully considered each of your points and revised the manuscript accordingly. Alongside the revised manuscript, we provide the following responses to your comments.
>
> # Clarification on Type of Uncertainty Dealt in this Work (Sec. 3.2 & 3.3)
>
> We thank the reviewer for the insightful comment regarding the type of uncertainty considered in our setting. We have revised Sec. 3.3 accordingly to incorporate the requested clarifications.
>
> In addition, we further revised the section to provide a clearer explanation of the notations. Specifically, we compute the element-wise standard deviation across conditional samples and take its $\ell_2$ norm to obtain a scalar uncertainty value for each frame $t$.
>
> Regarding computational efficiency, we address this point in the subsequent section in response to the question on the quantitative analysis of compute cost.
>
> # Clarification on Experiments (Sec. 4)
>
> **Comparison Against PDE-Refiner.** Thank you for the comment. As you correctly pointed out—and as noted in Sec. 4.2—PDE-Refiner serves as the base model for our approach. We would like to clarify that PDE-Refiner with $s=1$ and $s=6$ do not necessarily represent the upper and lower performance bounds. This is also reflected in our experiments on Gray–Scott, Cahn–Hilliard, and Anisotropic Diffusion, where the best performance is achieved with $s=1$, $s=3$, and $s=5$, respectively. This observation motivated our work, which aims to improve predictive accuracy through adaptive stepping while incurring only  minimal additional cost.
>
> **Quantitative Analysis of Computational Efficiency.** While our work primarily focuses on addressing error accumulation during autoregressive rollouts, we appreciate the suggestion and agree that reporting efficiency-related metrics would be informative. Since our method is based on PDE-Refiner, the NFE of PDE-Refiner and our method is identical, provided that all particles used for uncertainty estimation fit into GPU memory and can be processed in a single batch. In addition, we verified that the computations required for uncertainty estimation, such as computing the standard deviation across particles, do not introduce a significant bottleneck, which is consistent with the runtime breakdown shown in Tab. 5 of the revised manuscript.
>
> **Question Regarding Normalization of Sampling Procedure for Compute.** Could you elaborate on your question regarding the “normalization of the sampling procedure for compute”? While we may not be able to fully address this within the response period, we would be happy to incorporate any necessary clarifications or revisions in a future update. In the meantime, we note that when generating samples with PDE-Refiner and with our method, we do not impose a fixed compute budget across different step-size choices. As you correctly noted and as shown in the table below, PDE-Refiner with $s=6$ is approximately six times faster than PDE-Refiner with $s=1$.
>
> **Details of Temporal Derivative-Based Uncertainty Estimation.** Thank you for the question. We have revised Sec. 4.4 and provided a detailed explanation of the temporal derivative-based uncertainty estimation method.
>
> **Revising Table 3 to a Figure (Sec. 4.4).** We appreciate the suggestion and have replaced Tab. 3 with a figure in the revised manuscript (see Fig. 5). We also included data points from PDE-Refiner using constant step sizes $s=1,\dots,6$ in the figure.
>
> **Discussion on $s_{\text{avg}}$.** As noted in our earlier response in 'Comparison Against PDE-Refiner', please note that PDE-Refiner is the baseline where we always select the same step size $s=1, \dots, 6$ during rollouts. In Fig. 5 of the revised manuscript, we find that relaxing the threshold to $\tau=3$, results in higher $s_{\text{avg}}$ of approximately 2, at the cost of slightly increased predictive error.
>
> To better understand how adaptive stepping behaves, we included an additional paragraph and Fig.6 discussing the behavior of our method on the Gray–Scott and Cahn–Hilliard datasets.
>
> In accordance with our clarification on 'Comparison Against PDE-Refiner' and as shown in the quantitative comparisons in Tab. 1 of the manuscript, we would like to note that using a fixed step size of $s=1$ does not necessarily yield the best performance. This observation motivates our work, which aims to adaptively select step sizes, thereby improving predictive accuracy.
> Consistently, the behavior illustrated in Fig. 6 shows that our method selects varying step sizes across trajectories, with the resulting $s_{\text{avg}}$ aligning closely with the step sizes of the best-performing PDE-Refiner variants on each dataset.

---

> > ### Comment · Reviewer_U8mi · 2025-12-17
> > **Response**
> >
> > My apologies for me taking a while to reply! Thanks for taking the time to reply to my comments and to update the manuscript.
> >
> > In Figure 6, we see that at the average step size is smaller at the beginning of the trajectory than at the end. Would you have some intuition as why that is? As there little error had time to accumulate at the very beginning I was thinking that one could do bigger jumps than at the end, which seems to not be true.
> >
> > I find Figure 5 really insightful. Would you know why even for PDE-refiner s=2 yield lower relative L2 error than s=1? for your proposed approach is even starker that s=1 is worse that the s_avg for $\tau=1$. I was thinking that these curves would be monotonically increasing with $s$ which is apparently not the case. In some way that is the only way for DiffusionRollout to over perform PDE-refiner with an s_avg very close to 1, by 'skipping' some steps that would lead to error accumulation, but very unclear to me why and how.
> >
> > Looking at Table 5, I guess DiffusionRollout with $\tau$ does not bring any significant speedup compared to PDE-refiner with $s=1$. In some way though, perhaps it would be fairer to compare  PDE-refiner with $s=2$ since that achieves the best performance vs DiffusionRollout with $\tau=7$ since that seems to achieve the same performance, which would showcase that DiffusionRollout can yield speedups for same performance.
> > Although I am not certain how fair it is to say that sampling $K=2$ is as fast as $K=1$ as long as they fit in a single batch since, a) even though typically complexity is sub linear in $K$ it is still slower than $K=1$ and b) for some problem one cannot fit more than $K=1$ in a single batch. It is definitely often the case that one is interested in several samples which would then bring 'for free' this uncertainty estimate, but still worth noting this.

---

> > > ### Author Response · Authors · 2025-12-22
> > > **Author Response**
> > >
> > > Dear Reviewer U8mi,
> > >
> > > We sincerely appreciate you taking the time to review our revised manuscript.
> > > We have summarized our responses to your queries in the following.
> > >
> > > **Smaller Step Sizes in Early Trajectory Stages (Fig. 6)**
> > > Based on our observations, we attribute this behavior to be dataset-specific. In particular, Gray–Scott simulations exhibit rapid state changes in the early stages, followed by convergence toward steady states as time progresses. Consequently, uncertainty is higher at the beginning of the rollout. Our model therefore automatically selects step sizes that are better suited to each scenario, balancing the trade-off between network approximation error and condition-induced error.
> > >
> > > **Discussion on the Superior Performance of PDE-Refiner for $s \neq 1$**
> > > Thank you for your comment and for finding the provided figure useful. As you pointed out, taking smaller steps does not necessarily lead to better performance. Our intuition is that, consistent with the discussion of network approximation error and condition-induced error in Sec. 3.3, network approximation errors accumulate in this case. Therefore, decreasing the step size, and thus increasing the number of function evaluations (NFEs), can actually degrade accuracy.
> > >
> > > **Comments on Runtime Comparison with PDE-Refiner**
> > > Thank you for this insightful comment. As you correctly pointed and shown in Fig. 5, DiffusionRollout with $\tau = 7$ achieves error comparable to PDE-Refiner ($s = 2$), while using an average step size of approximately 2.5. This result indicates that adaptive stepping enables faster inference without sacrificing accuracy.

---

### Decision · Action_Editor_4tkS · 2026-01-02

**Recommendation:** Accept as is

**Audience:**

Yes

**Audience Explanation:**

Surrogate PDE modelling is an important topic in the scientific ML community. This paper proposes and justifies a method that will be of interest to this community, especially since it provides across-the-board accuracy improvements over existing methods.

**Claims And Evidence:**

Yes

**Claims Explanation:**

This paper hypothesize that tuning the step size of generative/neural PDEs solvers based on the uncertainty of probabilistic autoregressive rollouts can improve the accuracy of these solvers. The authors justify this hypothesis in Section 3 (Figure 2). The authors offer a comprehensive evaluation on several long-trajectory benchmarks with useful tables and ablations. The paper is structured well and easy to read.